

# The influence of green human resource management on university sustainability in higher education: the role of mediating environmental performance and green commitment

Aida Ahmed Zahrani

Department of Business Administration, College of Business Administration, Majmaah University, Al Majmaah, Saudi Arabia

## ABSTRACT

The purpose of the research is to examine how green human resource management (GHRM) contributes to the sustainability of the university. This study also focuses on how environmental performance and dedication to sustainability can act as mediators. Additionally, it seeks to examine the interplay between GHRM practice and university sustainability and how green environmental success and green dedication moderate that interaction. As the primary technique of data collection, a survey questionnaire on green HRM and environmental factors was distributed to a total of 273 university staff at Majmaah University in Saudi Arabia. The results of staff environmental performance and commitment point to a beneficial impact on sustainability in higher education institutions. Our study also demonstrates that when an employee scores highly on green performance evaluation and appraisal, the impact of green HRM practices on green dedication is more significant. By connecting green HRM practices to university sustainability through sustainability impact and green commitment, the current study adds fresh theoretical insights into the field of environmental management literature. Our findings give staff members advice on how and when to use green HRM techniques to improve university sustainability in higher education institutions.

# INTRODUCTION

These days, the most talked-about themes are green organizations and environmental sustainability (*Amjad et al., 2021*; *Pham, Tučková & Chiappetta Jabbour, 2019*). Due to its advantages, such as improving environmental performance, stimulating new ideas, motivating employees to commit to green tasks, and raising employee commitment levels in relation to the environment in businesses (*Siyambalapitiya, Zhang & Liu, 2018*; *Murshed, Abbass & Rashid, 2021*), the notion of GHRM has been organized all over the world. Human resources management (HRM) experts need to rethink the mission as well as extend the reach of their practices through the assimilation of green management practices to improve

Corresponding author
Aida Ahmed Zahrani,
aida.z@mu.edu.sa

how they carry out the fundamental HRM practices, as organizations are currently shifting their techniques and priorities toward more environmentally friendly agendas (*Del Brio, Junquera & Ordiz, 2008*).

According to *Ansari, Farrukh & Raza (2021)*, HRM can measure and have an impact on employee motivation, awareness, and behaviors connected to sustainability. As a result, businesses may effectively create and develop environmentally friendly policies using HRM (*Aboramadan, 2022*). An increasing number of colleges and universities all over the world have recently made an effort to integrate green initiatives and green practices into the services they offer. Higher education institutions are anticipated to play a vital role in implementing strategies and alternatives to address current environmental concerns as teaching and research institutions (*Lozano et al., 2019*; *Leon-Fernandez & Domínguez-Vilches, 2015*; *Disterheft et al., 2012*). Additionally, they must establish a precedent by reviving and acknowledging the changing demands and difficulties of environmental protection concerns (*Menon & Suresh, 2020*; *Gilal et al., 2019*). The teaching, research, and administrative staff at higher education institutions must implement green and ecologically friendly techniques in their daily work operations (*Anwar et al., 2020*). Green behaviors are generally defined as employee actions that support environmental management strategies at work (*Dumont, Shen & Deng, 2017*). Green workplace practices are seen to be most successfully implemented when employee green behaviors are taken into account. Additionally, studies have shown that educating employees on green practices is essential for environmental management projects (*Amjad et al., 2021*; *Mazzi et al., 2016*), as this will improve environmental performance and give employers a competitive edge (*Kim et al., 2019*).

Green human resources management (GHRM) practices are regarded as a crucial HRM technique to increase employees' environmental awareness at work in order to drive green employees' behavior. Green hiring, green training, green rewarding, and green performance evaluation are only a few of the functions included in GHRM processes designed to promote environmental management (*Dumont, Shen & Deng, 2017*; *Chaudhary, 2020*). Additionally, educational institutions are thought to be the biggest consumers of energy and resources due to the expansion of instructional activities and excessive usage of IT equipment (*Alshuwaikhat & Abubakar, 2008*) and (*Usman et al., 2022a*). Universities are accountable for the environment and incorporate environmental management (EM) elements and activities into their strategies, development and research plans, operational tasks, information technology, and course curricula (*Huang et al., 2022*; *Yusliza et al., 2019*). A few studies have examined employees' commitment to environmental duties, which is helpful to comprehend the connection between ecological dedication and HRM (*Amjad et al., 2021*; *Pham, Tučková & Chiappetta Jabbour, 2019*; *Tairu, 2018*; *Luu, 2018*). Contrary to key studies that concentrated on the effect of environmentally friendly HRM on corporate EP and employees' pro-environmental behavior, prior research has yet to identify the impact of green HRM practices on green commitment. Additionally, regarding the ability motivational opportunity (AMO) theory, *Blumberg & Pringle (1982)* suggested a framework to investigate the effects of green HRM practices like hiring, training, and evaluation; however, the researcher only found a small number of studies in the literature

that demonstrate the impact of green HRM on environmental commitment and behavior. In recent years, GHRM research has expanded, with studies conducted in various sectors, including technology and information (*Luu, 2018*; *Ojo & Raman, 2019*), and (*Chaudhary, 2019*). However, there is little study on GHRM in higher education (*Fawehinmi et al., 2020*; *Gilal et al., 2019*). One of the few studies found that GHRM increases academic employees' green behaviors by mediating the role of ecological knowledge, while another found that it is essential to incorporate employees' green behaviors into higher education organizations' management doctrine in order to enhance organizational financial and ecological performance and to win over employees.

Therefore, this study aims to develop a model illustrating how GHRM practices impact environmental performance and green commitment, specifically within the context of university sustainability. The objective is to advance the field of green HRM, both generally and in higher education. The study proposes that environmental performance and green commitment act as mediators in these relationships. Given the nascent stage of the connection between GHRM and organizational sustainability, this research contributes by expanding general knowledge on GHRM and addressing gaps in research within higher education (*Pham, Hoang & Phan, 2020*; *Bin et al., 2019*; *Yong, Yusliza & Fawehinmi, 2020*; *Aboramadan et al., 2020*; *Ren, Tang & Jackson, 2018*). Moreover, the study introduces a novel model that incorporates six additional elements into the existing GHRM literature: environmental orientation, green training and development, green recruitment and selection, green motivation, green performance management and appraisal, and top management support. Lastly, the research aims to deepen understanding of the mechanisms linking GHRM with environmental performance and foster sustainability commitments within universities. As a result, the current study uses GHRM practice as a theoretical framework to address the following research topics in the context of higher education in a developing nation: RQ1. Do GHRM practices encourage sustainability at universities? RQ2 what part do environmental performance and a commitment to sustainability play in a university's sustainability?

## RESEARCH MODEL AND HYPOTHESES

Understanding green HRM and environmental performance while considering the ability-motivation-opportunity theory (AMO) is crucial. Previous studies indicate that this approach provides the best insight into how GHRM practices influence an organization's performance (*Harrell-Cook, 2000*; *Boselie, Dietz & Boon, 2005*).

According to theory, high-performance activities are linked to HRM procedures that are gathered based on three factors: opportunities, opportunities for growth, and motivations (*Harrell-Cook, 2000*). Ability comprises participation in hiring training, which attests to the abilities and knowledge required by resources to carry out specific jobs. Opportunity, on the other hand, comprises information exchange, which motivates employees to engage in various activities. The final component of motivation is environmental orientation, which improves resource performance in order to accomplish organizational goals (*Marin-Garcia & Tomas, 2016*; see Fig. 1).

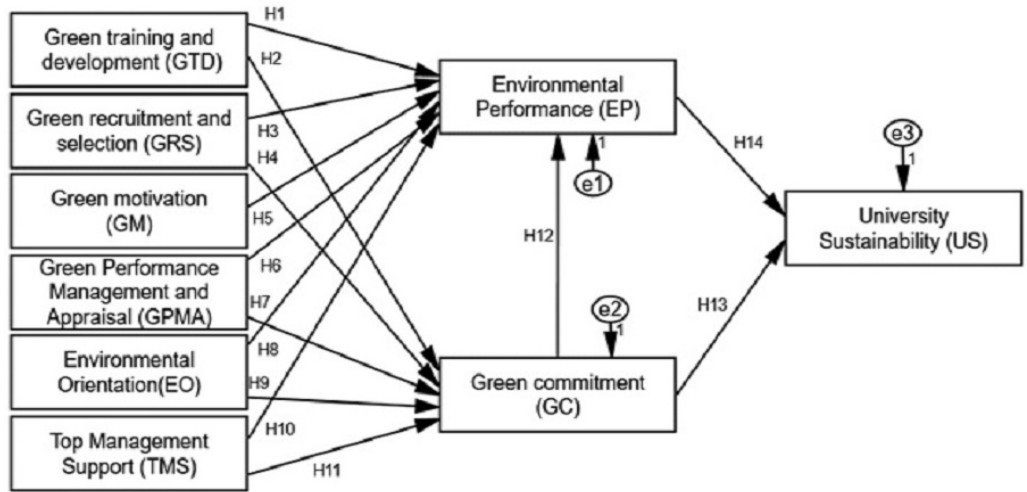

**Figure 1 Conceptual model.**

## Green training and development

Various training programs are provided by organizations that are designed to fulfill the goals of sustainability. This will improve employee training in effective green practices, such as how to lessen or stop the production of greenhouse gases, as well as managerial and technical skills in safeguarding natural resources (*Masri & Jaaron, 2017*; *Cheng et al., 2022*). Organizations may fairly easily see the benefits of green training and development programs for organizational and environmental sustainability (*Aboramadan & Karatepe, 2021*). The greatest issues facing businesses today are how to continue their economic growth while upholding high environmental sustainability (*Pinzone et al., 2019*; *Amjad et al., 2021*). In addition to these benefits, green training may be able to increase environmental awareness and establish positive attitudes and behaviors in both management and non-managerial personnel (*Pinzone et al., 2019*; *Alavi & Aghakhani, 2021*). GHRM is a special type of human resources planning that ensures sustainability on all levels (monetary, social, and ecological) (*Ababneh, 2021*). Then, the following theories were put forth:

*H1. Environmental performance benefits from green learning and development.*

*H2. Green commitment is boosted by green learning and development.*

## Green recruitment and selection

Candidates who care about the environment at work can be hired by businesses (*Jabbour, Santos & Nagano, 2008*). According to *Siyambalapitiya, Zhang & Liu (2018)*, in order to attract the best talent, green recruitment techniques' environmental regulations are expected to have a significant impact on attracting workers who are environmentally conscious. Due to increased understanding of the climate, the ecological reputation and identity of a recruiting agency are also crucial factors in recruitment (*Renwick, Redman & Maguire, 2013*). The environmental accomplishments of a corporation might be exploited to pique attention in the procurement stage (*Jabbour, Santos & Nagano, 2010*). Compared to more traditional channels like newspaper advertisements or brochures, web-based

recruitment enables recruiters to provide more information about their environmental protection efforts (*Renwick, Redman & Maguire, 2013*). Therefore, it is possible to assume:

*H3. Environmental performance benefits from green recruitment and selection.*

*H4. Green commitment is boosted by green recruiting and selection.*

## Green motivation

Green performance appraisal and incentive systems are part of green motivation (*Renwick, Redman & Maguire, 2013*). By encouraging managers to adopt green behaviors and evaluating their staff members based on green goals, HR can practice environmental leadership (leadership focus) (*Pham, Tučková & Chiappetta Jabbour, 2019*). The green goals ought to be established in accordance with environmental concerns (leadership emphasis). For the performance management process to be successful, management must effectively communicate (message credibility) about the key indicators and targets. Additionally, the HR department should collaborate closely with the other department heads to foster peer involvement and employee empowerment by offering rewards for putting out green initiative implementation. A compensation contingent that rewards improvements in green performance (*Kivinda et al., 2021*) can be used to accomplish this. Employees that reach their green goals may receive extra compensation (*Daily & Huang, 2001*). The HR division is in a better position to explain, disseminate, and transmit information (message credibility) about how such a compensation system functions within the company (*Van Waeyenberg & Decramer, 2018*). The HR department can also use this as an opportunity to spread environmentally friendly messaging that addresses employee concerns about reducing pollution levels while doing daily duties (*Van Waeyenberg & Decramer, 2018*). HR can promote a peer involvement system through performance appraisal and reward systems (*Pellegrini, Rizzi & Frey, 2018*). More specifically, key performance indicators (KPIs) are created in collaboration (peer engagement) between the heads of HR and other departments, with an emphasis on pro-environmental programs. These KPIs might be based on reduced resource usage, waste reduction, and recycling (*Ndubisi, Al-Shuridah & Capel, 2020*). In order to execute green initiatives, HR promotes peer involvement (workers collaborating with their peers) (*Van Waeyenberg & Decramer, 2018*; *Pellegrini, Rizzi & Frey, 2018*).

*H5. Environmental performance benefits from green selection and incentive.*

## Green performance management and appraisal

Employees may be encouraged to embrace green practices within the organization using green performance measurement and appraisal systems (*Jackson et al., 2011*). The performance assessment (PA) considers environmental responsibilities, such as fixing environmental problems and rules like reducing carbon emissions. Organizations should use corporate-wide measures to assess resource acquisition in order to encourage higher staff performance (*Tulsi & Ji, 2020*). Employee participation in green activities will increase if management compensates workers depending on their environmental performance evaluation (*Jabbour & De Sousa Jabbour, 2016*). Additionally, managers ought to encourage staff members to voice their concerns about the workplace and their own roles

in implementing green HR practices. Future goals that are geared toward implementing these environmentally friendly ideas and measuring employee progress should be developed by management. *Sharma and Gupta (2015)* assert that assessments of green performance are frequently focused on the traits of green productivity (*Ahmad & Allen, 2015*). It is suggested that HR unions should enhance employee assessments by allowing workers to be judged on both conduct and specialized knowledge in environmental protection. The following hypotheses can be used to more formally express these relationships:

*H6. Environmental performance benefits from green performance evaluation and appraisal.*

*H7. Green commitment is positively impacted by measurement and performance management and evaluation.*

## Environmental orientation

The recognition and implementation of moral values as environmental duties by an organization is referred to as environmental orientation (*Luu, 2020*). The phrase "corporate state of mind", which is used to describe environmental orientation, states that all business processes can affect and influence it (*Nair & Ndubisi, 2013*). According to *Luu (2020)*, there are environmental orientations that are both internal and external. Management and employees who establish and recognize values that support the need for environmental protection are included in the internal environmental orientation. On the other hand, an organization's interactions with stakeholders are part of its external environmental orientation (*Mansoor et al., 2021a*). An earlier study found that the presence of a well-integrated environmental system will increase the precision of the basis toward environmental protection (*Nazir et al., 2019*). Furthermore, it has been claimed that the adoption of an effective EMS that takes business initiatives and policies into account will lead to enhanced environmental performance (*Islam et al., 2020*). The efficiency of the environment will subsequently improve with the development of an ecological corporate culture that incorporates environmental ideals across the entire company (*Islam et al., 2020*). As a result, the following assertion was made:

*H8. Environmental performance is positively impacted by environmental orientation.*

*H9. Environmental Orientation enhances a person's commitment to being green.*

## Top management support

Numerous organizational and environmental elements have been put up by researchers as predictors of the adoption of green practices (*Abdel-Maksoud, Kamel & Elbanna, 2016*). Levels of environmental unpredictability, regulatory pressure, and customer pressure are a few examples of environmental factors. On the other hand, organizational characteristics include things like corporate size or support (*Lin & Ho, 2011*). With a few exceptions, these aspects haven't been extensively taken into account in research on green HRM, despite being taken into account in earlier studies looking at green practices (*Guerci, Longoni & Luzzini, 2016*). By looking at significant organizational factors that influence the adoption of green HRM in higher education, this study helps to close this research gap. These are environmental orientation and top management support. The adoption of green practices in firms has been credited with having top management backing (*Daily & Huang,*

*2001*). In particular, how senior management teams perceive environmental concerns as opportunities or dangers affects environmental initiatives at the corporate level (*Mansoor et al., 2021b*). It is clear that tighter integration between environmental concerns and business activities has become ever more crucial in high-risk industries like the oil and gas industry in order to save company costs and boost its reputation (*Mansoor et al., 2021b*). *Luu (2020)* contend that exploiting sustainability impact as a source of competitive advantages helps businesses find new business prospects by taking environmental challenges into account at the strategic level.

*H10. Environmental performance benefits from top management support.*

*H11. Support from top management influences green commitment favourably.*

## Green commitment

When applied to the environment, commitment reveals how engaged and accountable employees are to environmental issues as well as their level of internal motivation (*Luu, 2018*). Regular employee involvement in environmentally friendly initiatives improves their concerns about corporate environmental plans; as a result, employee dedication is high, which is useful in resolving institutions' environmental issues (*Usman et al., 2022b*; *Jabbour, Santos & Nagano, 2008*). According to experts, focusing on creating the EM structure can improve the green attitude of employees who are committed to the environment (*Raineri & Paillé, 2016*). As a result, their philosophy criteria improved to align with businesses' green goals and values, and now they choose to work with green businesses (*Pinzone et al., 2019*). Environmental performance is substantially impacted by management commitment (*Amir, Rehman & Khan, 2020*). Further investigation revealed that environmental performance is positively impacted by staff dedication (*Pham, Hoang & Phan, 2020*). Performance in the hotel sector is improved by managers' environmental responsibilities (*Tariq, Yasir & Majid, 2020*). The performance of an organization is greatly impacted by its environmental commitment (*Tariq, Yasir & Majid, 2020*). The following theories were put forth based on the literature:

*H12. Environmental performance benefits from commitment to sustainability.*

*H13. University Sustainability Benefits from Green Commitment.*

## Environmental performance

According to the definition of environmental performance, it is "the assessment of an organization's ability to achieve ecological goals and objectives that have been established in accordance with the organization's plan or policy." In order to achieve a competitive advantage, some businesses are currently implementing environmentally friendly projects (*Fraj, Matute & Melero, 2015*). The importance of ecological performance is thought to be a good opportunity to improve organizational success in a win-win scenario. Organizations from all over the world have been focusing on various green activities and how they affect the environment over the past several years (*Shen, Dumont & Deng, 2018*). Green practices are required since their adoption by enterprises can lead to higher performance (*Ali et al., 2022*). A few researchers illustrated how GHRM and green manufacturing might be prepared simultaneously to achieve environmental performance (EP) (*Singh et al., 2020*).

They also came to the conclusion that GHRM, which encourages workers to carry out their tasks in accordance with green standards, is the best strategy for achieving environmental performance. The following theories were put forth based on the literature:

*H14. Environmental performance influences university sustainability in a favourable way.*

## Sustainability

Organizations are responding to the urge to support sustainability more and more. Businesses are more likely to rely on their division of human resource management, a key internal resource, to implement a sustainability vision in order to accomplish this goal (*Wirtenberg et al., 2007*). In order to deal with different levels of pressure from governmental and non-governmental organizations like institutions, evolutionary developments, organizational renewal, and organizational effectiveness (*Bombiak & Marciniuk-Kluska, 2018*), HRM is crucial. As a result, a manager in the HRM department is likely to assign a specific focus to bringing about change and enhancing the business's sustainability initiatives (*Gim et al., 2022*). The company's personnel are viewed as a priceless resource since they carry out the organization's vision and mission (*Kim et al., 2019*). Therefore, the present research study uses RBV as the underlying theory to build and evaluate the empirical proof of the link in order to add to an ongoing discussion of the role of sustainable HRM practices in bringing organizational appropriateness. Few studies, however, suggest that GHRM is one of the practical instruments a company may use to generate organizational strategies that can aid in implementing sustainability practices (*Renwick et al., 2016*).

## RESEARCH METHODOLOGY

Our study evaluated a questionnaire sample while taking the advice of two experts. The green HRM component of the research model that was chosen encourages sustainable higher education among staff members in educational institutions. A 5-point Likert scale was used to evaluate all the collected data, including demographic and green HRM components. The questionnaire was physically distributed, and all respondents were asked to complete written versions in order to receive input on Green HRM for Environment Performance and Green Committed as well as their thoughts on how it will affect university sustainability. Data were gathered, and IBM SPSS and structural equation modeling were used to assess them (SEM-Amos). IBM SPSS and SEM-Amos are regarded as the main statistical methods employed in our study, which consists of two stages. The first stage involved developing the measures' discriminant, convergent, and convergent validity, and the second stage examined the structural model. This approach was recommended by *Hair et al. (2012)* and *Alzahrani, Stahl & Prior (2012)*. The research study mentioned above involved the collection of data from Saudi Arabia, and prior ethical approval was duly obtained under Reference No. Ethical Approval (MUREC-Jan-02/COM-2023/23-4)/Dated: 02-01-2023 and under research project no. Majmaah University for Research Ethics committee (MUREC)- H-01-R-088.

**Table 1  Demographic attributes.**

|  |  | Frequency | % |
|---|---|---|---|
| Gender | Male | 163 | 59.7 |
|  | Female | 110 | 40.3 |
| Age | 20–25 years | 16 | 5.9 |
|  | 26–30 years | 49 | 17.9 |
|  | 31–35 years | 142 | 52.0 |
|  | 36 and above years | 66 | 24.2 |
| Educational level | Bachelor's level | 3 | 1.1 |
|  | Master's level | 9 | 3.3 |
|  | PhD level | 261 | 95.6 |
| Work experience | Less than 1 | 15 | 5.5 |
|  | 1–3 years | 23 | 8.4 |
|  | 4–6 years | 110 | 40.3 |
|  | Above 7 years | 125 | 45.8 |

## Participants

Nearly 290 questionnaires were distributed, with a high return rate of 94.1%, resulting in 273 completed responses. After manual assessment, 17 incomplete questionnaires were excluded. The remaining 273 surveys were entered into SPSS for analysis. These surveys were distributed to staff members at Majmaah University in Saudi Arabia during August and September 2022, all of which proved to be usable. According to *Hair et al. (2012)*, the sample size for SEM should be determined by several factors, including the complexity of the model, the number of indicators per factor, and the desired level of statistical power. According to a commonly cited rule of thumb, the sample size should be at least 5 to 10 times the number of parameters under estimation. Based on *Hair et al.*'s (*2012*) guidelines the minimum sample size should be five times the number of parameters. $5 \times 45 = 225$ for better reliability, the preferred sample size should be 10 times the number of parameters ($10 \times 45 = 450$). Given that the study's sample size is 290, this number is adequate and falls within the acceptable range (between 5 and 10 times the number of parameters), ensuring robust statistical power and reliable results.

Out of 273 topic samples, 110 (40.3%) of the respondents were female and 163 (59.7%) of the respondents were male. From this poll, 16 (4.9%) respondents were between the ages of 20 and 25; 49 (17.9%) respondents were between the ages of 26 and 30; 142 (52.0%) respondents were between the ages of 31 and 35; and 66 (24.2%) respondents were older than 36. According to respondents' educational levels, three (1.1%) were bachelor's degree holders, nine (3.3%) were master's degree holders, and 261 (95.6%) were PhD holders. Demographic factors of work experience: 15 of the respondents were less than 1 year old with a percentage of 5.5%; 23 of the respondents were 1–3 years old with a percentage of 8.4%; 110 of the respondents were 4–6 years old with a percentage of 40.3%; and 125 of the respondents were above 7 years old with a percentage of 45.8%. See Table 1.

## Measurement instruments

With minor revisions and extreme care to preserve the original meaning, validated scale items for evaluating the dependent, dependent, mediating, and moderating factors were taken from past research. All constructs were measured using five-point Likert scales; the lowest score was 1 (strongly disagree), and the highest was 5 (Strongly agree). Five items were used to quantify the estimated "green training and development (GTD)," each of which was borrowed from (*Yong et al., 2020*; *Malik et al., 2020*; *Ren, Tang & Jackson, 2018*). Five (5) items were used to rate the "green recruitment and selection" factor; five (5) items were used to rate the "green motivation (GM)" factor; five (5) items were used to rate the "green motivation" factor; five (5) items were used to rate the "green motivation" factor; and five (5) items were used to rate the "green performance appraisal and appraisal" factor. Five (5) elements were used to rate the estimated "environmental orientation" factor, and (*Obeidat, Al Bakri & Elbanna, 2020*) and (*Ren, Tang & Jackson, 2018*) accepted each one. "Top management support" was also taken into account in more than five (5) items, and (*Obeidat, Al Bakri & Elbanna, 2020*) came in second. Additionally, *Roscoe et al. (2019)* graded the estimated "environmental performance factor" across five (5) different criteria. More than five (5) items also took into account "green commitment," which was followed by *Abbas et al. (2022)*. Last but not least, five (5) criteria were used to quantify university sustainability, as stated by *Abbas et al. (2022)* and *Obeidat, Al Bakri & Elbanna (2020)*.

## RESULT AND ANALYSIS

With a Cronbach's reliability (CR) coefficient of 0.926, the linked parameters had an impact on employees' environmental performance and commitment in higher education while taking organizational sustainability into account. Three criteria were used in this study to evaluate discriminant validity: variable index values below 0.80 (*Hair et al., 2012*), average variance extracted (AVE) scores higher than inter-construct correlations (IC) linked with factors, and AVE values considered equal to or greater than 0.5 (*Fornell & Larcker, 1981*). Additionally, the items and crematory factors of the construct's examination provided factor loadings of 0.7 or more, which were deemed to be acceptable by Cronbach's alpha and a composite reliability value of 0.70 or higher (*Hair et al., 2012*).

### Measurement model analysis

The main statistical tool employed in this study to analyze the results is confirmatory factor analysis (CFA) in AMOS 23. Structural equation modeling (SEM) with AMOS is a powerful statistical technique used for testing and estimating causal relationships using a combination of statistical data and theoretical causal assumptions. It integrates factor analysis and path analysis, allowing researchers to examine complex relationships among observed and latent variables simultaneously. Convergent validity, unidimensionality, consistency, and discriminant validity were all examined in this model. *Hair et al. (2012)* also suggested that model evaluation should be conducted using the highest likelihood estimation procedure and goodness-of-fit strategies, these include the chi-square statistic, which assesses the difference between the observed covariance matrix and the model-implied covariance matrix. A non-significant chi-square value ($p > 0.05$) suggests that

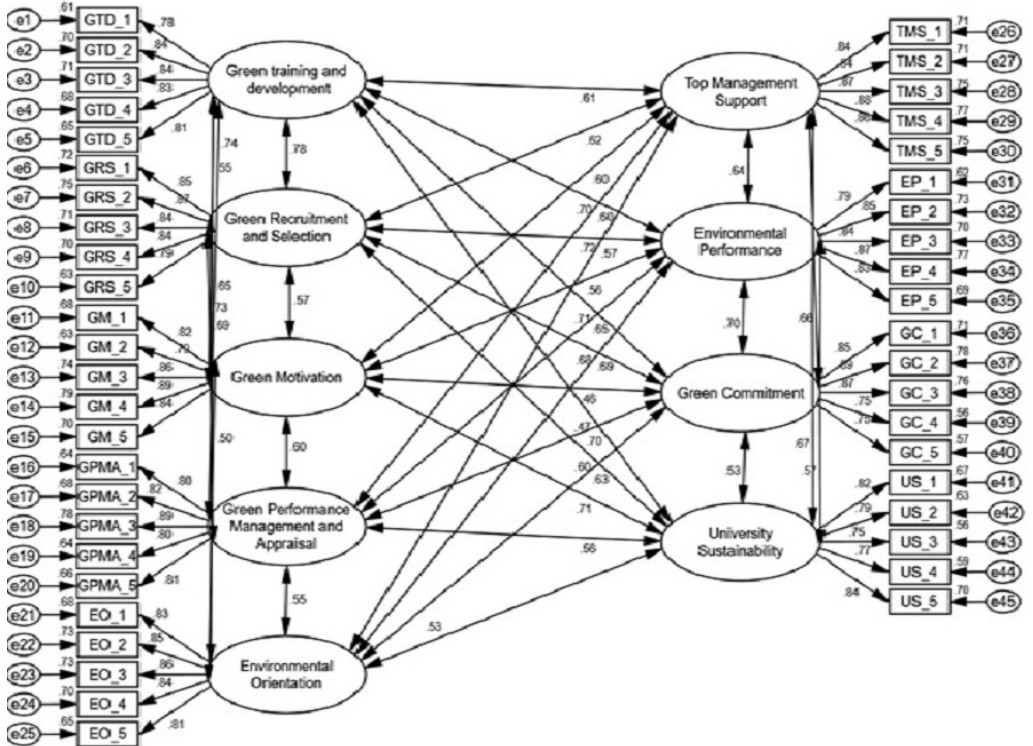

**Figure 2  Measurement of independent, mediator and dependent variables.**

the model fits the data well. Additionally, normed fit indices like the Normed Fit Index (NFI), which adjusts chi-square for degrees of freedom, indicate good fit with values close to 1. The Tucker-Lewis Index (TLI) and Comparative Fit Index (CFI) compare hypothesized models with baseline models; values closer to 1 indicate better fit, typically above 0.90 for acceptable fit. The Incremental Fit Index (IFI) compares hypothesized and independent models, also aiming for values above 0.90 for good fit. Root mean square error of approximation (RMSEA) measures model-data discrepancy, with values below 0.08 indicating good fit, which are taken into consideration by *Hair et al. (2012)*. Figure 2 measures independent, mediator, and dependent variables, and Table 2 summarizes the goodness-of-fit indices used to evaluate the models.

## Validity and reliability

Discriminant validity evaluates the degree of perception while taking into account different concepts' markers (*Bagozzi, Yi & Nassen, 1998*). The resulting AVE values showed that all of the values were greater than 0, indicating that all of the tested constructs agreed upon discriminant validity (*Fornell & Larcker, 1981*). Furthermore, according to *Hair et al. (2012)*, correlations between any two constructs cannot be greater than the square root of the variance that each item contributes to each construct on average. Additionally, the derived composite dependability values are shown, and they distinctly fall within the acceptable range of 0.70 and above. In addition, the Cronbach's alpha readings were in the

**Table 2** Summary out of goodness-of-fit indices.

| Measure type | Acceptable level of fit | Values |
|---|---|---|
| Chi-square (x2) | 3.5 to 0 (perfect fit) and ($p > .01$) | 1,993.104 |
| Normed Chi-square (x2) | Value should be >1.0 and <5.0 | 2.219 |
| Incremental Fit Index (IFI) | Value should be = or >0.90 | 0.913 |
| Tucker Lewis Index (TLI) | Value should be = or >0.90 | 0.901 |
| Comparative Fit Index (CFI ) | Value should be = or >0.90 | 0.913 |
| Root-Mean Square Error of Approximation (RMSEA) | Value <0.10 means a good fit and <0.05 indicates very good fit | 0.46 |

**Table 3** Validity and reliability for overall.

| | GTD | TMS | GM | EO | GPMA | GRS | GC | EP | US | CR | AVE | CA |
|---|---|---|---|---|---|---|---|---|---|---|---|---|
| GTD | 0.786 | | | | | | | | | 0.910 | 0.670 | 0.910 |
| TMS | 0.483 | 0.630 | | | | | | | | 0.932 | 0.734 | 0.933 |
| GM | 0.441 | 0.523 | 0.728 | | | | | | | 0.919 | 0.694 | 0.923 |
| EO | 0.475 | 0.458 | 0.415 | 0.805 | | | | | | 0.921 | 0.700 | 0.920 |
| GPMA | 0.536 | 0.482 | 0.478 | 0.410 | 0.794 | | | | | 0.907 | 0.663 | 0.913 |
| GRS | 0.576 | 0.494 | 0.464 | 0.513 | 0.533 | 0.794 | | | | 0.920 | 0.696 | 0.920 |
| GC | 0.532 | 0.557 | 0.411 | 0.496 | 0.539 | 0.519 | 0.848 | | | 0.909 | 0.667 | 0.911 |
| EP | 0.500 | 0.504 | 0.453 | 0.502 | 0.470 | 0.522 | 0.530 | 0.767 | | 0.921 | 0.700 | 0.920 |
| US | 0.341 | 0.522 | 0.551 | 0.383 | 0.402 | 0.438 | 0.408 | 0.409 | 0.766 | 0.896 | 0.633 | 0.893 |

range of the suggested value of 0.70 and higher. Additionally, readings for average variance extracted (AVE) fell between the acceptable value of 0.50 and higher. This shows that the overall loading factor is significant and exceeds 0.50, matching the criteria outlined in *Hair et al. (2012)* and *Fornell & Larcker (1981)* references. The following sections display the measurement model's data that was obtained; see Table 3.

## Structural model and hypotheses

As mentioned earlier, the proposed hypotheses were analyzed using confirmatory factor analysis (CFA), which confirmed the discriminant validity of Cronbach's Alpha (CA), average variance extracted (AVE), and Cronbach's reliability (CR) values, all of which met acceptable criteria. Table 3 displays all data for both male and female respondents. Additionally, the achieved composite dependability values are shown, and it is evident that they all fall above the threshold of 0.70, ranging from 0.896 to 0.932. Additionally, the Cronbach's Alpha values all exceeded the cut-off value of 0.70, ranging from 0.893 to 0.923. Additionally, AVE values were all higher than the suggested limit of 0.50, ranging from 0.633 to 0.734. This suggests that full factor loadings were important and greater than 0, matching the requirements (*Hair et al., 2012*; *Fornell & Larcker, 1981*). Only two out of the twelve hypotheses involving the nine main constructs were rejected in the current
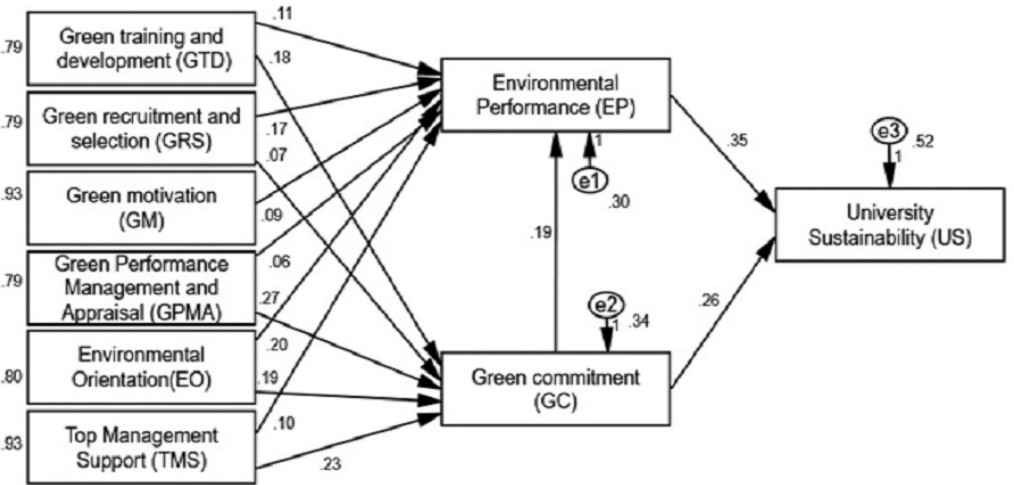

**Figure 3 Results for the proposed model.**

**Table 4 Hypothesis testing results.**

| Label | Factors | Relationship | Factors | Estimate | S.E. | C.R. | P |
|-------|---------|--------------|---------|----------|------|------|---|
| H1 | GTD | —> | EP | 0.110 | 0.039 | 2.807 | 0.005 |
| H2 | GTD | —> | GC | 0.182 | 0.040 | 4.549 | 0.000 |
| H3 | GRS | —> | PE | 0.171 | 0.038 | 4.541 | 0.000 |
| H4 | GRS | —> | GC | 0.074 | 0.040 | 1.860 | 0.063 |
| H5 | GM | —> | EP | 0.092 | 0.035 | 2.647 | 0.008 |
| H6 | GPMA | —> | EP | 0.055 | 0.040 | 1.365 | 0.172 |
| H7 | GPMA | —> | GC | 0.266 | 0.040 | 6.651 | 0.000 |
| H8 | EO | —> | EP | 0.201 | 0.039 | 5.161 | 0.000 |
| H9 | EO | —> | GC | 0.194 | 0.040 | 4.887 | 0.000 |
| H10 | TMS | —> | EP | 0.102 | 0.037 | 2.739 | 0.006 |
| H11 | TMS | —> | GC | 0.231 | 0.037 | 6.272 | 0.000 |
| H12 | GC | —> | EP | 0.188 | 0.057 | 3.300 | 0.000 |
| H13 | GC | —> | US | 0.260 | 0.066 | 3.921 | 0.000 |
| H14 | EP | —> | US | 0.354 | 0.071 | 4.963 | 0.000 |

sample, indicating the grouping of employees as shown in Fig. 3. The gathered information validated both the study model and the put forth hypothesis (Fig. 3).

All of the hypotheses are true, as indicated in Table 4 and Fig. 3, with the exception of two, which is "no green selection and recruitment between workers for greens commitment". Additionally, there is no green performance appraisal or staff evaluation for sustainability impact in sustainable universities. The current sample demonstrates that staff groups do engage in peer evaluation and green performance management, which results in green dedication for green HRM in higher education (0.266-H7).

As a result, each construct's hypothesis was stronger than that of the other constructions. When especially in comparison with its other theory value (*e.g.*, green training and

development (GTD) to environmental performance (EP) (H1: $\beta = 0.110$, $t = 2.807$), the Theory of Green Performance Management and Appraisal (GPMA) was shown to be significantly and positively related to green commitment (GC) for green HRM in higher education ($\beta = 0.266$, $t = 6.651$). The hypothesis that environmental performance (EP) is favorably and significantly associated with school sustainability (H14: $\beta = 0.354$, $t = 4.963$) is another example. While the relationship between green motivation (GM) and environmental performance (EP) shows the lowest hypothesis value (H5 $\beta = 0.092$, $t = 2.647$).

## DISCUSSION AND IMPLICATIONS

This study looked into how GHRM practices affected sustainability in the higher education sector in Saudi Arabia. It specifically looked at how organizational interventions such as green training, development, recruitment, and selection, motivation, performance appraisal and evaluation, enviro alignment, top management support, sustainability practices, and commitment affected desired university outcomes, such as the sustainability of the university. This investigation has made new ground by addressing a topic that has received little attention in the GHRM literature (*Shen, Dumont & Deng, 2018*) and (*Mousa & Othman, 2020*): how GHRM practices affect sustainability. According to our knowledge, this study is the only one of its kind to examine the connection between green HRM practices and university sustainable development in the context of higher education. It does this by examining the relationship among green learning and development, green hiring and selection, green inspiration, green performance appraisal and appraisal, environmental orientation, and top management support. By examining the impact of green HRM practices on university sustainability in higher education settings, our research specifically contributes to the expansion of earlier studies on environmentally friendly HRM practices with exogenous factors and environmental management. Second, the current study is advancing the investigation into whether environmental performance and environmental commitment act as mediators for the effects of environmental training and development, green hiring and selection, green inspiration, green performance appraisal and appraisal, environmental orientation, and support from top management on environmental university sustainability.

According to our data, environmental performance and commitment to the environment are positively correlated with the universities' green HRM practices, which is consistent with expectations. This can be due to the fact that changing employees' attitudes and behaviors can enhance university sustainability. These findings are in line with those from other studies (*Zahrani, 2022*; *Yafi, Tehseen & Haider, 2021*), which indicate that eco-friendly HRM practices encourage employees to engage in eco-friendly behavior and aid in the development of environmentally conscious minds. Our statistics thus demonstrate that the university can improve its environmental performance and dedication to the environment by engaging in sustainable behavior. The findings indicate a strong relationship between green team innovation and green hiring and selection. This outcome is in line with research from wealthy nations like Saudi Arabia, Indonesia, the UK, and the USA (*Zahrani, 2022*;

*Anwar et al., 2020*; *Faisal & Naushad, 2020*). H1 and H2 were approved. The findings also highlight the importance of the connection between environmental performance (PE), green commitment (GC), and green training and development (GTD) for the long-term viability of universities. This result is in line with earlier research findings (*Yong et al., 2020*; *Malik et al., 2020*); the H1 and H2 hypotheses were also accepted. The findings corroborate the hypothesis that environmental performance is closely connected with green recruitment and selection (GRS), which have previously been supported by research *Masri & Jaaron (2017)* and *Ren, Tang & Jackson (2018)*. (H3). As a result, this investigation's third hypothesis was also accepted. The current analysis, however, shows that neither green commitment nor green selection and recruitment (H4) significantly affect sustainability. The sustainability of institutions is not significantly impacted by green hiring and selection. This may be due to the rarity with which environmental considerations are included in green hiring, selection, and commitment (*Masri & Jaaron, 2017*; *Ren, Tang & Jackson, 2018*).

Furthermore, the findings corroborate the theories that environmental performance is closely connected with green motivation (GM), which have been supported by past studies (*Aboramadan, 2022*) and (*Ren, Tang & Jackson, 2018*). (H5). As a result, this investigation's third hypothesis was also accepted (*Muisyo et al., 2022*). According to the results of the current study, employee performance was not substantially correlated with green performance appraisals (H6). These results contradict earlier research that claimed green performance appraisals had a significant impact on organizational performance (*Ren, Tang & Jackson, 2018*). In a similar vein, this study discovered that evaluation of green performance (H7) was substantially related to dedication to the environment (GC). These results are in line with other research demonstrating that green performance evaluation significantly affects institutions' sustainability (*Abbas et al., 2022*). The hypotheses (H8, H9) propose that environment-oriented sustainability impact and green dedication are accurate determinants of green HRM in university processes, based on the outcomes of this study. Results showed that environmental alignment and green commitment had a substantial impact on environmental efficiency and green commitment in higher education (H8 and H9). These results are consistent with one researcher's findings that the goal of environmental orientation positively affects environmental performance and commitment to sustainability (*Obeidat, Al Bakri & Elbanna, 2020*) and (*Al-Tit, 2020*). Furthermore, it was established that top management support (TMS) (H10 and H11) was a crucial determinant of environmental productivity and green commitment to university sustainability. These results are consistent with one study's finding that the goal of support from top management (TMS) (H10 and H11) has a favorable effect on environmental performance and dedication to sustainability (*Obeidat, Al Bakri & Elbanna, 2020*).

The findings showed a strong correlation between environmental performance (H12) and green commitment. The findings confirm what has been stated in green HRM (*Amjad et al., 2021*; *Ren, Tang & Jackson, 2018*; *Muisyo et al., 2022*; *Abbas et al., 2022*), emphasizing that workplace resources act as a motivating factor to boost workers' productivity. Furthermore, it was discovered that green commitment (H13) has a favorable impact on universities' sustainability. This suggests that employees who are more committed

to the environment are more likely to engage in sincere and valuable interactions with their company, which would ultimately motivate them to demonstrate positive outcomes like green outcomes (*Abbas et al., 2022*). Finally, according to the results of this study, hypothesis (H14) suggests that green HRM in higher education is directly correlated with environmental performance and university sustainability. The outcomes demonstrated a considerable impact of environmental efficiency for green HRM in higher education (H14) on collegiate sustainability. These results are consistent with one study's finding that the goal of environmental efficiency has a favorable effect on a university's sustainability (*Roscoe et al., 2019*).

## Theoretical and practical

This study integrates the sometimes unrelated topics of sustainability and traditional management, and the authors give a thorough grasp of the sustainability of institutions. Previous studies have emphasized the value of GHRM practices in relation to greening an organization and improving employee performance (*Aboramadan, 2022*; *Abbas et al., 2022*; *Zahrani, 2022*). These practices include green professional development, green hiring and selection, green inspiration, green performance appraisal and appraisal, and green environmental orientation. Decision-makers must pay attention to how GHRM practices are still growing in underdeveloped nations. Although the current study was conducted locally in Saudi Arabia, the results are applicable globally, especially to developing nations (*Dumont, Shen & Deng, 2017*). This is because sustainable and green management have gained international attention. This study makes several contributions to the body of literature. First, it incorporates GHRM principles, environmental performance and commitment, green motivation, green top management support, and the sustainability of educational institutions in the context of developing nations. The study looks at how employee performance and GHRM practices relate to attaining sustainable university performance in higher education. Second, it looks into how environmental performance and a commitment to sustainability among GHRM practices contribute to university sustainability. These linkages have not been investigated together in studies before in a higher education setting.

This study offers helpful advice to academic staff on how to choose GHRM for the sustainability of the university. The study also helps staff members learn how to more effectively and positively encourage university staff members toward environmental issues. The significance of GHRM practices in Saudi higher education is the main topic of this work. As a result of incorporating the GHRM idea into the university's vision and mission statement, top human resources and management now have the additional duty of putting green practices into effect. Therefore, as represented in their working decisions, senior management should integrate environmental measures into business perspectives and image. This study can help and inspire university management to create and connect specific GHRM with sustainable strategic goals. This may encourage a greater level of staff commitment to using sustainable environmental practices for improved environmental performance. The study is helpful for environmental orientation, top management support, green performance appraisal and appraisal, green selection and selection, and green

development and training in academic institutions. It establishes a connection between GHRM and ecology, which is essential for Saudi higher education's triple bottom line. This study strongly suggests that while developing GHRM strategies, top leadership in higher education and GHRM professionals build fundamental organizational concepts and principles. An essential procedure to establish a state of user fit is to hire academic and administrative staff who share a commitment to environmental protection. Additionally, HRM professionals in higher education should set a good example by regularly holding seminars on green initiatives and using various communication tools at work to convey their environmental principles and values.

By evaluating candidates' awareness of and readiness to participate in green management practices, GHRM practitioners can also evaluate candidates' environmental principles during the interview process. Additionally, GHRM staff members can offer higher education staff proper environmental conservation training and coaching, which will help staff adhere to organizations' environmental policies and raise understanding of environmental management issues. Finally, GHRM officials in higher education may tie employee environmental performance to performance evaluation and reward programs. This could involve keeping track of how much paper is used over time and how many printing jobs are completed, both of which can be monitored using printing monitors.

## CONCLUSION AND FUTURE WORKS

In conclusion, this study's key finding is that the GHRM practices considerably and favorably predicted Saudi Arabia's higher education sector's sustainability. Sustainability in the environment is increasingly important to organizational operations. This study supports earlier findings and advances our knowledge of how the concept of GHRM promotes sustainability, which boosts universities' organizational productivity in terms of CSR. In order to achieve a high correlation between GHRM and organizational performance, sustainability and results are crucial. The current study also emphasizes how green commitment and environmental performance can considerably improve sustainable organizational behavior. Regarding its environmental contribution, this study shows higher education how to strengthen its sustainable position by implementing GHRM techniques. In conclusion, using a theoretical foundation to link GHRM practices to university sustainability has improved our comprehension of the worldwide movement toward the green movement in the setting of a developing nation. The limitations of this study offer opportunities for new research projects in the future. First, data gathered from one university's employees at one particular moment was used to investigate the model proposed. This involves asking managers to assess their staff members' environmental practices. Second, the generalizability of the findings may be constrained by the fact that the data were gathered from employees of three faculties of higher education institutions. As a result, future research might think about employing larger samples to replicate the study model. In order to analyze the model over time, future studies might take into account a longitudinal research approach. The study also looked at two mediators between the investigated links. Future research may take into account additional influencing

factors such as organizational identification, perceived organizational support for the environment, and green participation. To check for variations among these sectors, future studies may look into testing the model in various service sectors, such as higher education and non-profits.

### Funding
The Deanship of Scientific Research at Majmaah University supported this work under Project Number No. R-2022-297. The funders had no role in study design, data collection and analysis, decision to publish, or preparation of the manuscript.

### Grant Disclosures
The following grant information was disclosed by the author:
The Deanship of Scientific Research at Majmaah University: R-2022-297.

### Competing Interests
The authors declare there are no competing interests.

### Author Contributions
- Aida Ahmed Zahrani conceived and designed the experiments, performed the experiments, analyzed the data, prepared figures and/or tables, authored or reviewed drafts of the article, and approved the final draft.

### Ethics
The following information was supplied relating to ethical approvals (i.e., approving body and any reference numbers):

Ethical approval:

The research study involved the collection of data from Saudi Arabia, and prior ethical approval was duly obtained under Reference No. Ethical Approval (MUREC-Jan-02/COM-2023/23-4)/ Dated: 02-01-2023 and under research project no. Majmaah University for Research Ethics committee (MUREC)-H-01-R-088.

### Data Availability
The data is available in the Supplementary File.

### Supplemental Information
Supplemental information for this article can be found online at http://dx.doi.org/10.7717/peerj.17966#supplemental-information.

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
