# Peer review of "The influence of green human resource management on university sustainability in higher education: the role of mediating environmental performance and green commitment"

_PeerJ, doi:10.7717/peerj.17966_

## Round 0.1 · original submission · Major Revisions

Dear Authors,

Please revise and improve your manuscript as advised by the reviewers

Regards

Reviewer 1 ·

Basic reporting

I have gone through the MS entitled” The influence of green HRM on university sustainability in higher education: the role of mediating environmental performance and green commitment”. I have reservations about the said MS regarding its scope with PeerJ life and environmental Sciences. Please see my comments below;
There is room for improvement of English language in the MS. Data are presented in the form of required tables, figures etc. Raw data also shared. Hypotheses well stated.

Experimental design

The MS examined how green human resource management (GHRM) contributes to the sustainability of the university. Moreover, this MS focused on how environmental performance and dedication to sustainability can act as mediators. Additionally, it examined the interplay between GHRM practise and university sustainable and how green environmental success and green dedication moderate that interaction.
To my understanding the MS looks like a case study/case report and has less relevance to Aims and Scope of the journal under consideration. The MS may have its usefulness/significance in other fields of research; therefore, I suspect the suitability of the submitted MS in this journal so, I recommend to decline this submission.

Validity of the findings

Conclusions well drawn on the basis of research performed.

·

Basic reporting

Self-contained with relevant results to hypotheses.

Experimental design

Research question well defined, relevant & meaningful. It is stated how research fills an identified knowledge gap.

Validity of the findings

All underlying data have been provided; they are robust, statistically sound, & controlled.

Additional comments

The influence of green HRM on university sustainability in higher education: the role of mediating environmental performance and green commitment, Manuscript is well written however it is better if authors send this manuscript for English editing from English native proofreading organization for improvement. However, there are few mistakes and deficiencies that should be addressed before preceding it further. I have some minor suggestions.

• Title may be revise with addition of full form of HRM.
• Keyword should be arranged alphabetically.
• Introduction is very lengthy. Please remove irrelevant information.
• Please rewrite the objective for more clarity.
• All the headings of manuscript need to revise. Please follow the author guidelines.
• Result and Analysis
• Discussion and implications
• Conclusion and future works

Reviewer 3 ·

Basic reporting

In this article, the author investigated the effect on green human resources management (HRM) on the sustainability of the university. The author proposed 14 hypothesis and verified them using questionnaires and statistical analysis. Overall, the article demonstrates the importance and benefits of green HRM in high education using both theories and real data. There are some specific comments:

1. Some words and sentences are incomplete or incorrect in English. Specifically,
1. Line 30, "thru" is not a formal English word.
2. Line 99-100, it is not a complete English sentence.
3. Line 101-102, it is not a complete English sentence.
4. Line 112, "fulfil" should be "fulfill".
5. Line 273-278, these sentences are a bit redundant and need to be rephrased.
6. Line 284, should be "less than 1 year".
7. Line 346-347, this sentence should be rephrased. "As previously mentioned, using CFA, CA, AVE, and CR values to analyse the hypotheses were discovered to ..."
8. Line 350, use "CA" for "Cronbach's Alpha".
9. Line 355, it is not a complete English sentence.
2. Provide more details in Table legends. For example, for Table2, what do the goodness-of-fit indices measure? For Table 3, what is the meaning of overall? What are the meanings of the abbreviations? For Table 4, hypothesis testing for what? And the meanings of abbreviations and the highlight?
3. Break the "instruction" section to multiple paragraphs. It is hard to follow right now.
4. The paragraphs of "Discussion and implications" section are too long to digest. Separate them to multiple paragraphs and make them more logical.

Experimental design

1. Line 318, provide more details about the SEM-Amos model.
2. Line 320-324, provide more details about the definition of these statistics, like the formulas, and how they are used in this case. Currently, it is hard to evaluate the results.
3. Give more details on how to evaluate the moderating effect.
4. Why these 290 participants were selected to fill the questionnaire? Are there any inclusion-exclusion criteria?

Validity of the findings

1. Line 307, how was Cronbach's reliability coefficient computed?
2. Line 360, the study was conduced on university staff, why was it related to student groups?
3. In Table 4, what is the meaning of the "C.R" column? For hypothesis testing, what is the threshold for significance? Did the author consider multiple hypothesis testing correction?

Additional comments

1. I suggest not using abbreviations in the title.
2. Line 310, "AVE" is used before definition.
3. Line 357, the quotation mark is not closed.

Annotated reviews are not available for download in order to protect the identity of reviewers who chose to remain anonymous.

---

## Round 0.2 · accepted · Accept

Authors have improved their manuscript. It may be accepted for publication.

Reviewer 1 ·

Basic reporting

I have gone through the MS entitled” The Influence of Green Human Resource Management on University Sustainability in Higher Education: The Role of Mediating Environmental Performance and Green Commitment”. Please see my comments below;
Although authors have greatly improved the English and they claim that they have consulted native English speaker, there is still room for improvement of English language in the MS. MS looks a mix of UK & USA English. e.g., Support from top management influences green commitment favourably. Adopt US English for PEERJ preferably.

Title is still confusing. Data are presented in the form of required tables, figures etc. Raw data also shared. Hypotheses well stated. Referencing looks not according to PEERJ.

Experimental design

ok

Validity of the findings

Conclusions well drawn on the basis of research performed.

·

Basic reporting

Literature references, sufficient field background/context provided

Experimental design

Research question well defined, relevant & meaningful. It is stated how research fills an identified knowledge gap

Validity of the findings

All underlying data have been provided; they are robust, statistically sound, & controlled.

Additional comments

No Further comments

Reviewer 3 ·

Basic reporting

The revised manuscript is much more readable than before. The language issues are properly addressed.

Experimental design

No issues are found in experiemental design

Validity of the findings

The findings are valid and convincing